# Selective ablation of VIP interneurons in the rodent prefrontal cortex results in increased impulsivity

**Jessica A. Hatter** [1]*, **Michael M. Scott**[1,2]

1 Department of Pharmacology, University of Virginia, Charlottesville, Virginia, United States of America,
2 Department of Toxicology, Charles River Laboratories, Edinburgh, Scotland

* jh6hn@virginia.edu

## Abstract

It has been well-established that novelty-seeking and impulsivity are significant risk factors for the development of psychological disorders, including substance use disorder and behavioral addictions. While dysfunction in the prefrontal cortex is at the crux of these disorders, little is known at the cellular level about how alterations in neuron activity can drive changes in impulsivity and novelty seeking. We harnessed a cre-dependent caspase-3 ablation in both male and female mice to selectively ablate vasoactive intestinal peptide (VIP)-expressing interneurons in the prefrontal cortex to better explore how this microcircuit functions during specific behavioral tasks. Caspase-ablated animals had no changes in anxiety-like behaviors or hedonic food intake but had a specific increase in impulsive responding during longer trials in the three-choice serial reaction time test. Together, these data suggest a circuit-level mechanism in which VIP interneurons function as a gate to selectively respond during periods of high expectation.

## Introduction

In both humans and mice, the medial prefrontal cortex (mPFC) bidirectionally regulates goal-seeking behavior, such as novelty-seeking and impulsivity [1]. Various studies have demonstrated abnormal activity of the mPFC in patients with substance use disorder (SUD) or behavioral addictions (e.g., pathological gambling, kleptomania, binge eating, compulsive sexual behavior) [2–12]. Additionally, impulsivity and novelty-seeking are strongly associated with both the development and maintenance of SUD [13, 14], and even moderate the efficacy of SUD treatment [15–17]. While dysfunction in the mPFC may underpin conditions that are often characterized by alterations in reward pursuit in both males and females [18–20], very little is known about how specific neuronal populations in the mPFC regulate these behaviors.

Extra-cortical glutamatergic, serotonergic, and cholinergic inputs converge onto vasoactive intestinal peptide (VIP)-expressing interneurons in the mPFC [21–24], placing them at an ideal position to serve as a mediator between long-range inputs and local cortical processing. VIP interneurons are associated with behavioral modification, especially following reward presentation [25, 26], implicating their role in novelty-seeking and impulsivity phenotypes. By

**Data Availability Statement:** All raw image files are available from the Figshare database (DOI: 10.6084/m9.figshare.22299013). All raw data is available from the Figshare database (DOI: 10.6084/m9.figshare.22297150).

**Funding:** MS, 5RO1MH116694-04, National Institute of Health, https://www.nih.gov/, The funders had no role in study design, data collection and analysis, decision to publish, or preparation of the manuscript.

**Competing interests:** The authors have declared that no competing interests exist.

providing inhibitory input onto somatostatin (SST) interneurons that innervate pyramidal (PY) neurons, VIP interneurons provide indirect, disinhibitory input onto PY neurons. Because of their unique position to regulate novelty-seeking and impulsivity, through both their myriad of inputs as well as their local circuit control, it is of particular interest to better understand how VIP interneurons function in the control of behavior.

The rodent mPFC is composed of three primary subareas–the anterior cingulate cortex (ACC), the prelimbic cortex (PL), and the infralimbic cortex (IL). Studies have suggested these mPFC subregions have differential and specialized roles in behavior, including the control of social interaction, palatable food intake, and novel object investigatory behavior [27–31]. These roles result from the distinct projections that each mPFC subarea receives from various other brain areas, which primarily converge onto VIP interneurons [24, 32–34]. As such, we hypothesized that selective ablation of VIP interneurons in the IL would be sufficient to modulate both impulsive responding as measured by a three-choice serial reaction time task and novelty-seeking as measured by novel animal investigation. Using an adeno-associated virus (AAV) construct encoding a cre-dependent caspase-3, we were able to investigate whether VIP interneuron control is necessary in the modulation of behavior. VIP ablation in the mPFC led to a specific increase in impulsive responding during long-delay trials, with no non-specific effects on anxiety-like behaviors or food-related motivation, revealing a novel role of VIP neurons in the control of impulsive behavior.

## Results

### Selective ablation of VIP interneurons in the mPFC

Various studies have described the differential and specialized roles the mPFC subregions have in the control of social behavior. For example, Huang, *et al.* found that activation of projections from the PL to the basolateral amygdala (BLA) impaired social interaction, while inhibition of IL-BLA projections also impaired social interaction [27]. This is most likely due to the distinct projections that each subarea receives, as well as the differences in the areas that each subarea innervates [32–34]. Additional studies have directly implicated the PL and IL in behavioral inhibition, indicating that inhibition of PL neurons increased premature responses, while inhibition of IL neurons decreased premature responses in a response preparation task [29]. Because VIP neurons are the primary convergence point of projections into the mPFC, we hypothesized that VIP neurons would be an important gateway point for behavioral modification. Therefore, we aimed to resolve the contribution of IL VIP neurons to the control of impulsive behavior, as measured by a three-choice serial reaction time task (3CSRTT). To evaluate this contribution, we measured 3CSRTT reaction times in mice wherein IL VIP neurons were ablated. In order to visualize VIP neurons, we created a VIP::ZsGreen mouse line in which a VIP-cre drives a floxed ZsGreen reporter to specifically label VIP neurons. Ablation of the VIP interneurons was achieved via a bilateral injection of a cre-dependent Caspase 3 into the border of the IL and dorsal peduncular cortex (DP) of VIP::ZsGreen mice, thus causing apoptosis of cre-expressing VIP neurons (Fig 1A). Sham animals were given an identical injection of sterile saline. We confirmed VIP-specific ablation through the specific loss of ZsGreen-expressing neurons (unpaired t-test, p = 0.0164, Fig 1B). Cre-dependent ablation was also confirmed to be primarily contained to the IL region versus the PL region of the mPFC (S1 Fig).

### Ablation of VIP interneurons increases impulsive responding in long-delay trials

The 3CSRTT is designed to measure motor impulsivity as a characteristic of prefrontal cortex activity [36]. In this task, the mouse is trained to recognize an illuminated nose poke hole and

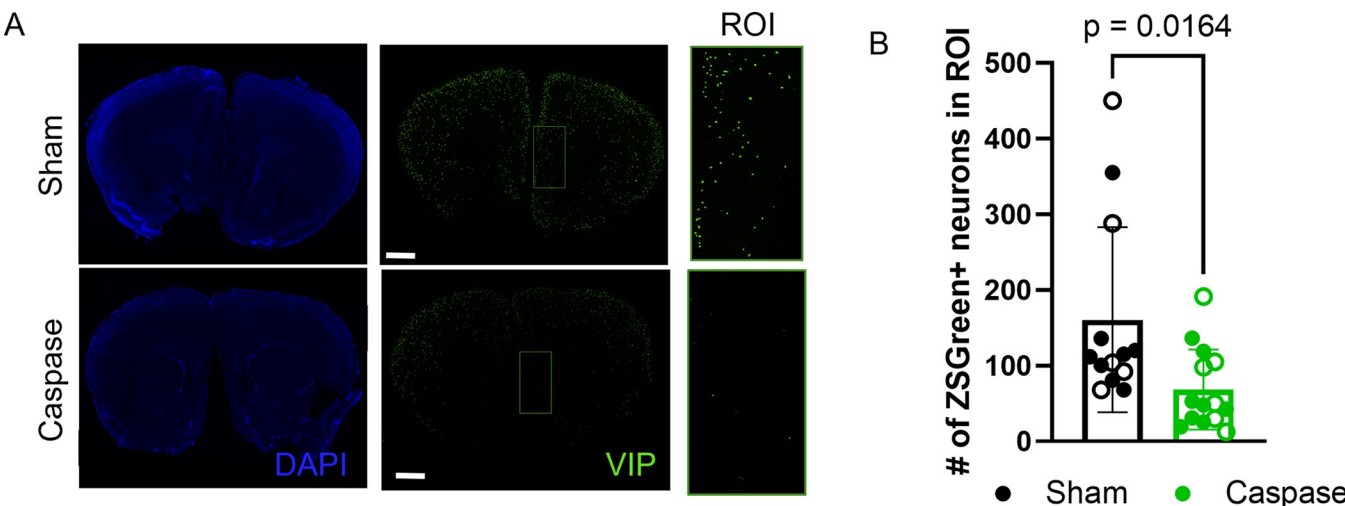

**Fig 1. Caspase ablation localized to the IL and DP.** (A) AAV-drive cre-dependent caspase ablation of VIP interneurons was successful in selectively ablating VIP interneurons, as indicated by loss of ZsGreen fluorescence. Scale bar: 1 mm. (B) Successful ablation of VIP interneurons was observed in all study animals (unpaired t-test, p = 0.0164), between +1.10 mm and +1.98 mm rostral of the bregma, corresponding to panels 14 to 22 in Paxinos and Franklin [35]. ROI was decided using pilot animals to determine spread of AAV and then centered around injection coordinates. Open circles = male, closed circles = female.

must poke within a 5 second window of time to receive a reward. Four distinct behaviors are measured during this task: (1) omission, in which the mouse does not respond to the cue within 5 seconds, (2) incorrect, in which the mouse pokes in the incorrect hole, (3), correct, in which the mouse pokes in the correct hole, and (4) premature, in which the mouse pokes during the intertrial interval (ITI) that occurs between the response time and the following cue. The number of premature responses serves as an indicator of impulsive action. We trained both VIP-ablated and sham mice in the 3CSRTT (Fig 2A). While there was not a statistically significant difference in premature, correct, or incorrect responding between sham and caspase-treated animals when all ITIs were sampled together, there was a trend towards ablated mice having an increased proportion of premature responses (unpaired t-test, p = 0.093, Fig 2B). Based on data indicating that the infralimbic area is responsible specifically for behavioral control of long-delay trials [29], we then separated trials by the ITI and found that ablated animals had a significant increase in premature responses exclusively when the ITI was set to 12.5 s (p = 0.004, Fig 2C). This observation was similar across both male and female populations, with no significant differences between the sexes (S2 Fig, S1 and S2 Tables).

## Ablation of VIP interneurons does not affect interest in novel animals

Previous studies demonstrate that activation of IL VIP neurons reduces novel animal investigation [31], implicating a role of IL VIP neurons in the control of novelty seeking behavior. To evaluate this role, we tested mice on a novel social interaction assay, as previously described [37]. In this assay, mice were given 150 s to explore an open field with two restrainers, and then 150 s to explore the open field with a novel mouse in one of the restrainers (Fig 3A). Mice that underwent VIP ablation did not spend significantly more time exploring the novel mouse (unpaired t test, p = 0.5532, 95% C.I. = [-24.79, 13.59]) and did not make first contact with the novel animal at significantly different times from the sham animals (unpaired t test, p = 0.5929, 95% C.I. = [-43.73, 25.52], Fig 3B). These findings were consistent in both male and female populations, with no significant changes between sexes (S3 Table, S3A Fig).

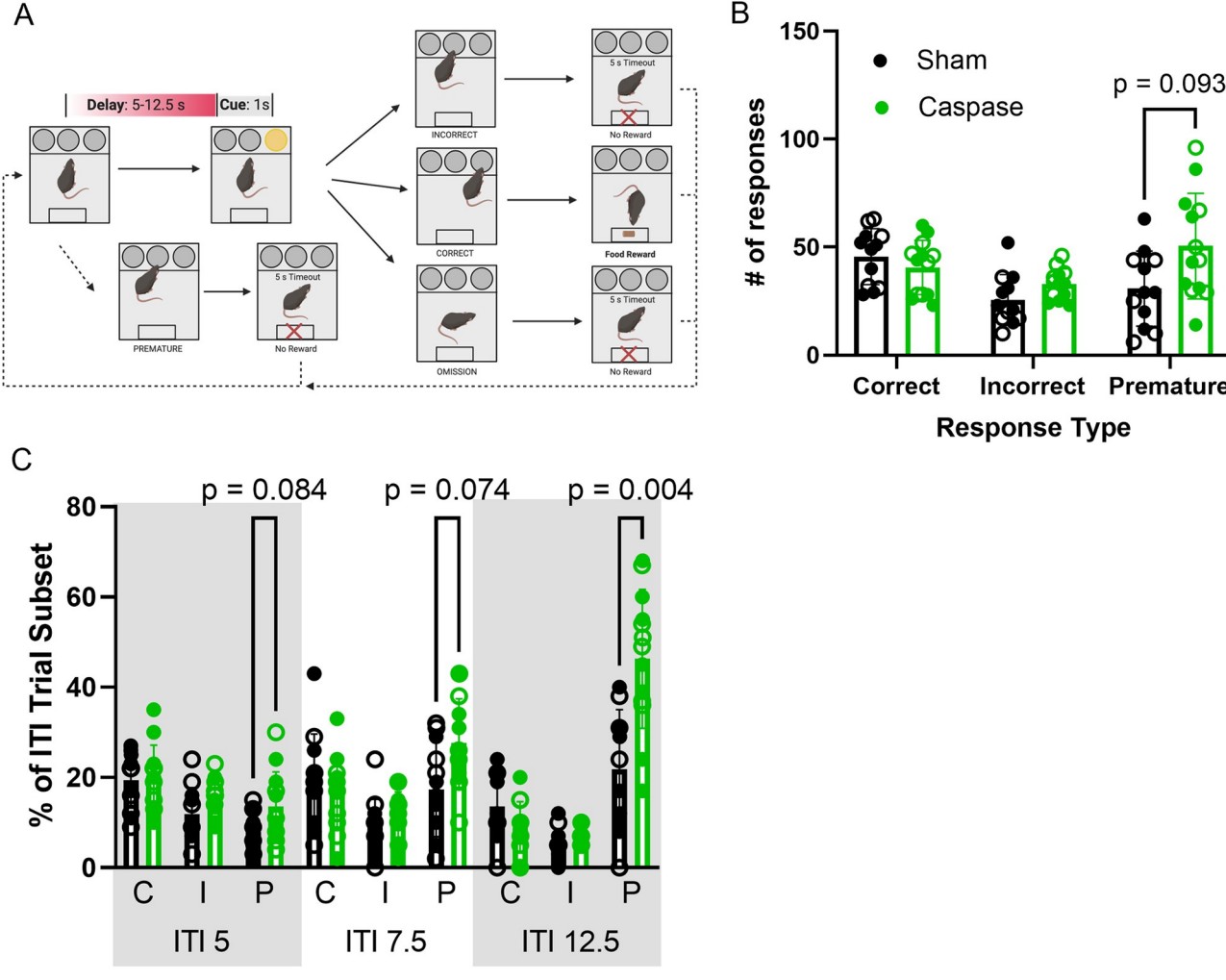

**Fig 2. Caspase ablation of IL VIP interneurons results in an increase in long-delay premature responses as measured by 3CSRTT.** (A) Task schematic of three-choice serial reaction time task. Created with Biorender.com. (B) VIP ablation in the IL results in a slight increase in premature responses across all inter-trial intervals. A two-way ANOVA revealed there was a statistically significant interaction between treatment and response type ($F(2,29) = 4.011$, $p = 0.0225$; unpaired t-tests, $p_{correct} = 0.344$, $p_{incorrect} = 0.108$, $p_{premature} = 0.093$). (C) When separated into discrete ITI categories, a two-way ANOVA demonstrates that there is a statistically significant interaction between treatment and ITI-dependent response type ($F(8,206) = 7.056$, $p < 0.0001$; unpaired t- tests, $p_{correct\ ITI5} = 0.662$, $p_{incorrect\ ITI5} = 0.112$, $p_{premature\ ITI5} = 0.084$, $p_{correct\ ITI7.5} = 0.503$, $p_{incorrect\ ITI7.5} = 0.303$, $p_{premature\ ITI7.5} = 0.074$, $p_{correct\ ITI12.5} = 0.133$, $p_{incorrect\ ITI12.5} = 0.303$, $p_{premature\ ITI12.5} = 0.004$). C = correct, I = Incorrect, P = premature. Open circles = male, closed circles = female.

However, caspase-ablated males showed a trend towards approaching a novel animal much faster than their sham counterparts (S3 Table, S3A Fig).

## VIP interneuron ablation does not increase spatial anxiety-like behavior

Because our behavioral tests revealed that VIP neuron ablation selectively increases impulsivity behavior (i.e., an increase in the number of impulsive responses as measured by the 3CSRTT), we tested the hypothesis that the observed increase in impulsivity may be a non-specific effect of an overall change in behavior. To assess this possibility, animals were subjected to an open field assay to determine any changes in spatial anxiety-like behavior. Spatial anxiety-like behavior was quantified by measuring the amount of time that an animal spent in the center of the open field, with the expectation that animals with higher levels of spatial anxiety will spend

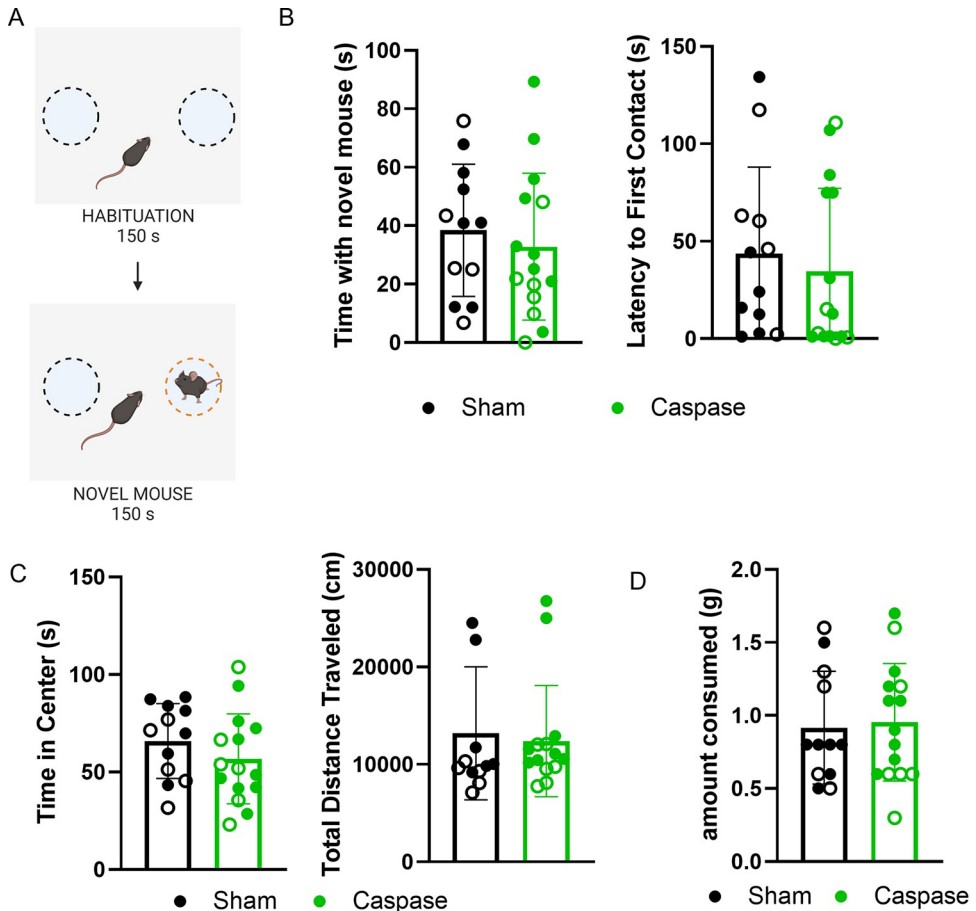

**Fig 3. Caspase ablation of IL VIP interneurons does not influence novelty-seeking or anxiety behaviors.** (A) Task schematic of novel social interaction assay. Created with Biorender.com. (B) VIP ablation in the IL does not affect time spent exploring a novel animal (unpaired t-test, p = 0.5532) or latency to approach a novel animal (unpaired t-test, p = 0.5929), (C) spatial anxiety-like behavior as measured by time spend in the center of an open field (unpaired t-test, p = 0.2851) and overall locomotion in an open field (unpaired t-test, p = 0.7447) or (D) amount of high-fat diet consumed (unpaired t-test, p = 0.8128). Open circles = male, closed circles = female.

less time in the center of the open field. VIP neuron ablation did not affect time spent in the center of the open field, suggesting that ablation does not increase anxiety-like behavior (unpaired t test, p = 0.2851, 95% C.I. = [-26.19, 8.036], Fig 3C). These findings were consistent across both male and female populations, with no differences between sexes (S3B Fig, S3 Table). Ablation of VIP neurons in the IL did not significantly affect overall locomotion when compared to sham animals (unpaired t test, p = 0.7447, 95% C.I. = [–5751, 4166], Fig 3C). A two-way ANOVA revealed that caspase ablation did not have a statistically significant effect on overall distance travelled, but sex was a statistically significant factor in the overall distance traveled (S3C Fig, S3 Table).

## Ablation of VIP interneurons does not increase palatable food intake

Because of the effect of VIP ablation on impulsive responding, it is possible that VIP ablation creates a general effect on novelty seeking, causing an increased motivation for non-chow food and a resulting increase in impulsive responses. Studies have demonstrated that optogenetic manipulation of the mPFC can alter free feeding [38–40], and that activation of the IL VIP

interneurons is sufficient to increase palatable food intake [31]. We therefore evaluated the effect of VIP neuron ablation of hedonic food intake. In this assay, mice are given a pre-weighed nugget of high-fat diet (Teklad TD.88137, 15.2% kcal from protein, 42.7% kcal from carbohydrate, and 42% kcal from fat) and allowed to eat freely for 30 minutes. We consider food intake during this time period, conducted just prior to the start of the light cycle, to be driven primarily by hedonic mechanisms, as our prior work has shown animals consume minimal amounts of food in this assay when tested with home cage dietary chow (Teklad 2013, 4% fat, 17% protein, 48% carbohydrate, no sucrose, 2.9 kcal/g) [31]. Ablation of VIP interneurons in the IL was not sufficient to produce a significant change in palatable food intake over a 30 min period (unpaired t test, p = 0.8128, 95% C.I. = [-0.2789, 0.3523, Fig 3D). These findings were consistent in both male and female animals, with no significant differences between sexes (S3 Fig, S3 Table).

## Discussion

The data presented here indicate that ablation of VIP interneurons in the IL is sufficient to drive impulsivity without increasing novelty-seeking or anxiety-like behaviors. We demonstrate here that VIP interneurons are necessary for control over impulsive responding specifically during long-delay trials. These behavioral changes occur without increasing anxiety-like behavior or palatable food intake, implicating VIP IL interneurons in the specific control of impulsive responding. To our knowledge, this marks the first behavioral exploration of IL-VIP interneuron ablation and further elucidates their role in the control of behavior.

Previous studies demonstrate that excitation of mPFC PY neurons with a Gq-coupled DREADD does not alter binge-like feeding or anxiety-like behavior but reduces impulsivity on the 3CRSTT task, but only after a high dose of CNO [37], consistent with our findings. In contrast, Hardung et al. found that optogenetic IL inhibition in rats suppresses early responses [29], which we did not find in our model of VIP ablation in mice. The differences in our findings likely result from differences in inhibition–while they chose to inhibit entire areas of the mPFC without distinction for neuron type, which would primarily target PY neurons, we have directly manipulated only VIP neurons. However, our findings indicate that VIP ablation in the IL only increased impulsive responding during long-delay trials, consistent with the findings of Hardung et al. insofar that the IL is implicated specifically in long-delay trials.

Though we found VIP ablation in the IL was sufficient to drive increased premature responding during long-delay trials, there is not sufficient power to determine if this phenomenon is sex-specific. We observed in S2 Fig. that there was a strong trend in both sexes towards increased premature responding in long-delay trials, but these remain non-significant based on our set statistical significance threshold ($p_{female}$ = 0.112, $p_{male}$ = 0.173). Additionally, while none of the behavioral assays indicate sex-specific differences, it is possible that these would become more apparent if our study had greater statistical power to detect these potential sex dependent effects.

While we have previously demonstrated that VIP stimulation via a cre-dependent stabilized step-function opsin (SSFO) expressed in VIP-cre animals was sufficient to suppress high calorie food intake and decrease overall locomotion, we found no effect on food intake during this experiment [31]. It is possible that this selective effect on animal behavior results is driven by the population of VIP interneurons that synapse directly onto PY neurons [21, 22]. Thus, direct activation of VIP interneurons would directly suppress novelty-seeking while ablation of the VIP interneurons could be compensated for by additional inhibitory output from parvalbumin (PV) and SST neurons. Additionally, recent research has shown that VIP neurons act as a type of gate, allowing us to also hypothesize that direct stimulation of the VIP neurons

is sufficient to "open" the gate, while ablation results in the gate continuing to remain "closed" and have no effect on novelty-seeking behavior [41].

A mechanism of adaptive disinhibitory gating would additionally explain why VIP-specific ablation had an effect only in long ITI trials. Krabbe, *et al.* found that VIP interneuron activation was strongly modulated by outcome expectations, revealing a novel form of disinhibitory gating in the control of learning and behavior [41]. We can therefore hypothesize that VIP interneurons are similarly affected by changing ITIs in our model of impulsivity and are activated specifically during longer periods of waiting. When VIP interneurons are ablated, this adaptive gate is absent, thus resulting in increased impulsive responding during longer ITIs. Finally, our findings are also consistent with the role that the mPFC plays in the control of appropriately timed reactions [42–44], as well as the role that the IL specifically plays in response inhibition [45–47], and sheds new light on the role of VIP interneurons in the control of the timing of this inhibition.

## Methods

### Experimental animals

All studies were approved by the University of Virginia's Animal Care and Use Committee. Twelve-week-old adult male and female VIP-IRES-Cre (VIP-Cre, Strain # 010908) and B6.Cg-Gt(ROSA)26Sor$^{tm6(CAG-ZsGreen1)Hze}$/J (Ai6, Strain # 007906) were purchased from The Jackson Laboratory. Ai6 contains a floxed STOP-cassette resulting in ZsGreen expression only in cre-expressing cells. Mice were housed in the Pinn Hall vivarium at the University of Virginia on a 12h light: 12h dark cycle (lights off at 21:00) with *ab libitum* access to food (Teklad 2013, 4% fat, 17% protein, 48% carbohydrate, no sucrose, 2.9 kcal/g) and water, unless otherwise stated. Both lines have been backcrossed to C57Bl6/j animals for at least 7 generations. We generated heterozygous VIP::ZsGreen animals through two subsequent crosses: (1) crossing VIP-cre homozygous females with Ai6 homozygous males and (2) crossing the resulting heterozygous VIP$^{cre/+}$/Ai6$^{fl/+}$ offspring (referred to as VIP::ZsGreen throughout). This strategy results in ZsGreen expression localized to VIP-expressing neurons. Animals (N = 27, $N^{F,sham}$ = 7, $N^{F,caspase}$ = 9, $N^{M,sham}$ = 5, $N^{M,caspase}$ = 6) were transferred to a flipped light cycle room (12h light: 12h dark, lights off at 10:00, on at 22:00) 1 week before surgery and throughout the remainder of behavioral experimentation to allow for all behavioral experiments to occur during Zeitgeber Time (ZT) 12-14hr, during a time of heightened animal activity and alertness. Animals were genotyped using the following primer sets: (1) for VIP-Cre: Mutant (Mut) Forward 5'– CCC CCT GAA CCT GAA ACA TA– 3', Common 5'–GCA CAC AGT AAG GGC ACA CA– 3', Wild Type (WT) Forward 5'–TCC TTG GAA CAT TCC TCA GC– 3' and (2) for Ai6: WT Forward 5' –AAG GGA GCT GCA GTG GAG TA– 3', WT Reverse 5'–CCG AAA ATC TGT GGG AAG TC– 3', Mut Reverse 5'–GGC ATT AAA GCA GCG TAT CC– 3', Mut Forward 5'–AAC CAG AAG TGG CAC CTG AC– 3'.

### Adeno-associated viral vector and stereotaxic viral injections

Mice were anesthetized with Ketaset (60 mg/kg, i.p., Zoetis, Parsippany, NJ, US) and Dexdomitor (0.45 mg/kg, i.p., Zoetis) and given Normasol (500 uL, s.c., Mölnlycke, Göteborg, Sweden), which we found to decrease surgical deaths. All injections were performed using Neurostar StereoDrive (Tübingen, Germany). VIP neurons were targeted using a Cre-dependent Caspase 3 virus, pAAV-flex-taCasp3-TEVp from Addgene [48]. We injected 400 nL of virus bilaterally into the boundary of the infralimbic cortex (IL) and the dorsal peduncular cortex (DP) of 8-10-week-old male and female VIP::ZsGreen mice using coordinates based on Franklin and Paxinos [35] (+1.54 mm from bregma, ±0.3 mm lateral of midline, and 3.3 mm

ventral of the dura). A Hamilton syringe fitted with a 26G needle was inserted to a depth of -3.3 mm and 400 nL of virus was delivered via pressure injection over a period of 12 minutes. To prevent delivery of the virus to more dorsal areas, the needle was left *in situ* for 10 minutes and then slowly removed. Control mice received sham surgery, wherein 400 nL of sterile saline was delivered bilaterally in the same manner as the AAV. Mice were given ketoprofen (5 mg/ kg, i.p.), antisedan (1mg/kg, i.p., Zoetis) and Normasol (500 uL, s.c.) to accelerate post-surgical awakening. After surgery, mice were singly housed throughout the duration of the experiments. After 14 days to allow for sufficient levels of viral vector expression and to allow the animals to fully recover from surgery, mice underwent behavioral assays.

## Brain tissue preparation

Mice were euthanized using 100 uL of Euthasol euthanasia solution (Virbac AH, Inc., Carros, France). Once mice no longer responded to a toe pinch, mice were first flushed with chilled phosphate-buffered saline, followed by perfusion with chilled 4% paraformaldehyde in 0.1-M phosphate buffer (4% PFA). Brains were kept in 4% PFA overnight and then transferred into 1X PBS until sectioning took place. Brains were dissected and sectioned at 40-μm thickness on a compresstome (Precisionary Instruments, Natick, MA, USA). Sections were mounted in sequential order, air-dried, and coverslipped in Vectashield hard-set mounting medium with DAPI (Vector Laboratories, Newark, CA, USA).

## Quantitative analysis of VIP ablation by caspase

Six 40 μm sections from +1.10 and +1.98 rostral of bregma were taken, corresponding to panels 14 to 22 in Paxinos and Franklin [35]. All slices were imaged at 4X magnification using an Olympus BX61 using manual tiling function. Neurons were counted within a pre-determined ROI from previous pilot experimentsusing ImageJ. All quantitative analysis of VIP ablation was performed in animals used in behavioral animals. Two animals (one sham female and one caspase-injected female) were removed from analysis because of non-specific expression of ZsGreen that was evident during analysis.

## Three-choice serial reaction time task

Behavioral assays were performed in the following order: (1) 3CSRTT, (2) open field test, (3) social interaction test, and (4) binge eating test. Behavioral training and testing occurred during the animal's dark cycle (ZT12-24) in a red-light lit behavioral room. Behavioral acquisition training is split into 13 stages and is performed in operant chambers (Med Associates, Inc, St. Albans, VT, USA). Briefly, in stage 1, all nose poke holes are illuminated and a nose poke in any hole results in reward delivery (Banana Flavor Pellets, #F06727, Bio-Serv, Flemington, NJ, USA). Subsequently, in stage 2, only the center hole is illuminated, and only pokes in this hole result in reward delivery. As training progresses, the duration of nose poke hole illumination is progressively reduced, ultimately reaching 0.5 seconds (s), and nose poke holes are illuminated in a pseudo-random order. During all stages of training and testing, mice must refrain from poking until the hole is illuminated and must wait 5 s to identify the correctly lit nose poke hole prior to poking. A premature poke or lack of response will result in a 5 s timeout and no reward. After completion of stage 13 of training, the intertrial interval (ITI, the time between the illumination of the nose poke holes) is lengthened to 7 s to produce a slight elevation in impulsive responding to allow improved data collection. During testing, the animals undergo 250 trials, in which the ITI is randomized between 5 s, 7.5 s, and 12.5 s in order to increase impulsive responding during longer ITI trials. Both training and testing are self-paced and

conducted over a 12-hour period each day, in *ad libitum* fed animals. A typical animal will finish training within 7–10 days (84 to 120 hours) and testing between 14–20 hours.

## Open field test

All behavioral testing occurred in a dedicated behavior room, separate from the home room, as conducted previously [31, 37]. The behavioral room is lit only by red light, allowing for minimal interruptions of the animals' circadian cycle during behavioral testing [49]. Two days after 3SCSRTT testing, mice were brought to the behavioral room and allowed to acclimate for 1 hour before testing began. Mice were placed into the PhenoTyper (Noldus, Wageningen, the Netherlands) and allowed to explore for 15 minutes while movement was recorded using Etho-Vision XT tracking software (Noldus). The PhenoTyper was cleaned between each mouse with Minncare disinfectant to remove residual odors. We waited 5 minutes between each animal to allow for any residual odor from the cleaning agent to dissipate. During testing, a yellow light was turned on in the PhenoTyper, to provide consistent illumination of the arena. To ensure that arena novelty was not a confounding variable during the social interaction assay, all mice underwent this experiment before all other experiments conducted in the PhenoTyper.

## Social interaction

The social interaction task was performed in the PhenoTyper, as previously described [31, 50]. Before the social interaction test, all mice were brought to the behavior room and allowed to acclimate for at least 1 hour. To allow for habituation, the chamber was first prepared with two empty restrainers on opposite sides of the PhenoTyper. The test mouse was placed in the center of the PhenoTyper and allowed to explore for 150 seconds. The test mouse was then returned to its home cage for 30 s while the restrainers were cleaned with Minncare and replaced. A novel mouse of the same sex was then placed in one restrainer and the test mouse was returned to the center of the chamber and allowed to explore for 150 s. The side of the chamber the novel mouse was placed on was randomized to minimize confounding variables due to lingering smells. The chamber was cleaned between each mouse with Minncare and allowed to air out for 5 minutes to remove residual odors. Mouse movement was recorded using Ethovision.

## Binge-like eating assay

Measurement of palatable food intake was performed as previously described [31, 51]. On the night before testing, mice received a small (<0.2 g) sample of the high calorie diet (Teklad TD.88137, 15.2% kcal from protein, 42.7% kcal from carbohydrate, and 42% kcal from fat, Envigo, Dublin, VA, USA), delivered into their home cage. At ZT 20:00, all food was removed, and mice were challenged with approximately 3g of pre-weighed high fat diet and allowed to consume freely. After 30 minutes, the food was removed and weighed, and mice were returned to *ab libitum* chow feeding conditions.

## Statistical analysis

All statistical analyses were performed in Prism 9 (GraphPad, Boston, MA, USA). Multiple comparisons were corrected for false positives using a false discovery rate correction (FDR) with a desired FDR set at 1.00%.

## Supporting information

**S1 Fig. Caspase ablation of VIP interneurons is primarily localized to the IL.** (A) Representative images of the spread of caspase-3 AAV. PL = prelimbic, IL = infralimbic, DP = dorsal peduncular cortex. (B) Quantitative analysis of ablation of VIP interneurons in the PL vs IL. Caspase ablation was localized to the IL, as indicated by a significant decrease of VIP interneurons (represented by ZsGreen expression) in the IL ($p_{IL} = 0.0254$) but not in the PL ($p_{PL} = 0.1092$). Additionally, there are significantly fewer VIP interneurons in the IL of caspase animals ($p_{caspase} = 0.0499$) but not in the sham animals ($p_{sham} = 0.8284$), indicating specific ablation of VIP interneurons in the IL.
(DOCX)

**S2 Fig. Caspase ablation of IL VIP interneurons does not cause discriminate impulsive behavior in males vs females.** (A) Caspase ablation results in a trend of increased premature responding in both male and female mice, with a stronger increase in female animals, while not significant. No significant differences between male and female animals. (B) ITI length corresponds to an increase in premature responses in both males and females but is not significant. No significant differences between male and female animals. Statistics summarized in S1 and S2 Tables.
(DOCX)

**S3 Fig. Caspase ablation of IL VIP interneurons does not cause discriminate novelty-seeking or anxiety-related behaviors in males vs. females.** (A) Caspase ablation does not affect novelty-seeking behavior as measured by novel social assay in either males or females. No significant differences between male and female animals. (B) Caspase ablation does not affect anxiety-related behavior as measured by time spent in center of open field and overall locomotion in either males or females. No significant differences between male and female animals. (C) Caspase ablation does not affect binge-like food intake in either males or females. No significant differences between male and female animals. Statistics summarized in S3 Table.
(DOCX)

**S1 Table. Statistics summary for S1A Fig.**
(DOCX)

**S2 Table. Statistics summary for S1B Fig.**
(DOCX)

**S3 Table. Statistics summary for S2 Fig.**
(DOCX)

## Author Contributions

**Conceptualization:** Jessica A. Hatter, Michael M. Scott.

**Data curation:** Jessica A. Hatter.

**Formal analysis:** Jessica A. Hatter.

**Funding acquisition:** Michael M. Scott.

**Investigation:** Jessica A. Hatter.

**Methodology:** Jessica A. Hatter.

**Writing – original draft:** Jessica A. Hatter.

**Writing – review & editing:** Jessica A. Hatter, Michael M. Scott.

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
