## [Decision Letter · Decision Letter 0]

1 Feb 2023

PONE-D-22-34113Selective ablation of VIP interneurons in the rodent prefrontal cortex results in increased impulsivityPLOS ONE

Dear Dr. Hatter,

Thank you for submitting your manuscript to PLOS ONE. After careful consideration, we feel that it has merit but does not fully meet PLOS ONE’s publication criteria as it currently stands. The two reviewers clearly outline their positive comments and their concerns with your study. Therefore, we invite you to submit a revised version of the manuscript that addresses the points raised during the review process.

We look forward to receiving your revised manuscript.

Kind regards,

Andrey E Ryabinin, Ph.D.

Academic Editor

PLOS ONE

Journal Requirements:

Reviewers' comments:

Reviewer's Responses to Questions

**Comments to the Author**

1. Is the manuscript technically sound, and do the data support the conclusions?

Reviewer #1: Yes

Reviewer #2: Partly

2. Has the statistical analysis been performed appropriately and rigorously? 

Reviewer #1: No

Reviewer #2: Yes

3. Have the authors made all data underlying the findings in their manuscript fully available?

Reviewer #1: Yes

Reviewer #2: Yes

4. Is the manuscript presented in an intelligible fashion and written in standard English?

Reviewer #1: Yes

Reviewer #2: Yes

5. Review Comments to the Author

Reviewer #1: The manuscript by Hatter and Scott characterizes the role of VIP-expressing interneurons in the infralimbic cortex. The authors used a cre-dependent caspase-3 approach to ablate VIP neurons in the IL. Using a three-choice serial reaction time task, the authors found an increase in premature responses in the caspase mice. The authors used both sexes in compliance with SABV. Overall, this discrete study adds to our understanding of the role of VIP interneurons in encoding impulsivity.

Major Issues:

1) Was the spread of the caspase restricted to IL? Please, show images of the spread of the virus. Given that the authors injected 400 nl, it’s most likely that the virus spread to PL.

2) The authors should reanalyze the data using the right statistics. Throughout the manuscript, the authors use unpaired t-tests post-ANOVA without correcting for multiple comparisons.

3) The correct way to test for sex differences is to run ANOVA with sex as a covariate. The authors should run 3-way ANOVA (Fig. S1) and 2-way ANOVA (Fig. S2) to correctly compare the effect of biological sex. If there is no main effect of sex or interaction between sex and treatment, only then should the authors claim no sex differences.

4) The number of mice used per sex/treatment is not powered enough to find sex differences. This limitation should be mentioned in the discussion.

5) In line 113, remove the statement “implicating a sex-specific control over novel social interactions” since there’s no statistical significance.

6) In prior work from the lab, Newmyer et al. found reduced locomotion following the activation of IL/PL VIP neurons. Did the authors observe any difference in the total distance traveled?

7) In Fig. S2A, there are 7 female sham and 9 female caspase mice, but in S2B, there seem to be 8 female sham and 10 female caspase mice. Same issue with S2C. Please fix this and report the correct values.

8) In Fig. S2, it looks like female caspase mice consumed more food than sham mice. Did the ablation of VIP neurons alter the body weight of mice?

Minor Issues:

1) Line 148 should say Gq-coupled DREADD

2) Line 156: it should be trials and not trails

3) Minor grammatical errors and spelling mistakes.

Reviewer #2: Hatter and Scott present a well-written manuscript assessing the behavioral effects of infralimbic cortex VIP ablation on impulsivity-like behavior and other mPFC-related tasks. They use a caspase virus with a transgenic strategy in VIP-IRES-Cre mice and combine male and female mice. VIP caspase-ablated mice showed increased impulsive-like behavior with the 12.5 intertrial interval, as assessed in the three-choice serial reaction time task (3CSRTT), with trends in the same direction at the shorter ITI’s and premature responses. There were no other stark differences between sham and VIP-ablated mice when they were further tested for social interaction with a novel mouse, free exploration in the Phenotyper box, and acute high-fat diet consumption. Importantly, these behaviors were performed during the dark phase. While this is a straight-forward set of studies with the same animals, improving the treatment of sex, increasing specificity of behavioral language, and including more methodological details would improve this manuscript.

These experiments collapse the data for male (n=5-6 per group) and female (n=7-9 per group) mice. While there are no significant differences in sex for any of the behaviors, this may be because of a low sample size, especially among the males. These behavioral differences may be driven by the females, as indicated in the Supplementary Table of p-values. If adding more animals is not possible, the authors could indicate the male vs female data points within the Figures. For example, having open vs closed circles, or circles versus triangles, etc. would distinguish the sexes within the same bar graph. Another recommended strategy would be to provide the means and standard errors of the variables in the supplementary tables (plus p-values, as already given) for each sex. This treatment of sex would not only improve the manuscript, but be useful for future studies.

I have some issue with anxiety-like behavior being assessed in the PhenoTyper. Traditionally the open field test is performed in larger, uncovered arenas, and the Noldus PhenoTyper is enclosed on all sides and a smaller area space. I would make it clear to readers that this set-up is not a traditional open field, reporting the smaller dimensions. I would also refrain from generalizing anxiety-like behavior since other tasks of anxiety-like behavior, such as an elevated plus maze, or light-dark test, were not performed. Again, I recommend being more specific with word choice.

Another area of overgeneralization is regarding the acute high fat diet feeding test. The authors present the experiment as assessing high calorie food intake, hedonic feeding, or eating behavior, but the protocol was a 30-minute test. Research on hedonic feeding may entail more chronic high fat protocols assessing binge-like consumption. Further, general eating behavior was not assessed, so adding more specific language, or discussing as a limitation, would more accurately reflect the results.

The Figures have poor resolution. In Figure 1A, one can hardly see the ROI box in the middle panels or see the text “DAPI.” In Figure 2A, the text labels are also illegible.

The methods are generally well-explained, but some minor methods details can be added, such as: 1) test order of experiments, 2) 3CSRTT training vs testing timeline, 3) equipment manufacturer for the 3CSRTT, 4) did mice encounter same-sex conspecifics.

Minor writing edits:

Abstract, The second to last line, “impulsive responding during longer trials” could end with “in the three-choice serial reaction time test.”

Line 103, The header, “Ablation of VIP interneurons does not affect interest in novel stimuli” is too general in that only a novel conspecific was tested, and not separate tests of social interaction and novel object recognition task.

Line 148, I think the phrase, “Gq-coupled does not alter” could use “DREADD” or “chemogenetic stimulation” in the sentence.

6. PLOS authors have the option to publish the peer review history of their article (what does this mean?). If published, this will include your full peer review and any attached files.

Reviewer #1: No

Reviewer #2: No

---

## [Author Response · Author response to Decision Letter 0]

18 Apr 2023

Re: Manuscript Resubmission to PLOS ONE [PONE-D-22-34113]

Our specific responses to each of the reviewers’ comments and questions are outlined below:

Journal Requirements:

We have accordingly updated the manuscript to fit PLOS ONE’s style requirements and updated the file names as required. 

2. In your Data Availability statement, you have not specified where the minimal data set underlying the results described in your manuscript can be found. PLOS defines a study's minimal data set as the underlying data used to reach the conclusions drawn in the manuscript and any additional data required to replicate the reported study findings in their entirety. All PLOS journals require that the minimal data set be made fully available. Upon re-submitting your revised manuscript, please upload your study’s minimal underlying data set as either Supporting Information files or to a stable, public repository and include the relevant URLs, DOIs, or accession numbers within your revised cover letter. 

We appreciate PLOS ONE’s dedication to making data publicly available and have included the DOI accession numbers here and in the cover letter as requested. 

DOI for raw image files: https://doi.org/10.6084/m9.figshare.22299013

DOI for raw data: https://doi.org/10.6084/m9.figshare.22297150

This statement has been removed from the manuscript as it was not a core part of the research being presented and had no bearing on the findings presented in the manuscript as written. 

Reviewer #1: The manuscript by Hatter and Scott characterizes the role of VIP-expressing interneurons in the infralimbic cortex. The authors used a cre-dependent caspase-3 approach to ablate VIP neurons in the IL. Using a three-choice serial reaction time task, the authors found an increase in premature responses in the caspase mice. The authors used both sexes in compliance with SABV. Overall, this discrete study adds to our understanding of the role of VIP interneurons in encoding impulsivity.

Major comments:

1. Was the spread of the caspase restricted to IL? Please, show images of the spread of the virus. Given that the authors injected 400 nl, it’s most likely that the virus spread to PL.

We agree that this information was lacking in our initial submission and Supplementary Figure 1 has been added to address this question. This figure now includes both representative images to show the spread of the caspase virus as well as quantitative analysis of PL and IL ZSGreen expression.

2. The authors should reanalyze the data using the right statistics. Throughout the manuscript, the authors use unpaired t-tests post-ANOVA without correcting for multiple comparisons.

We thank the reviewer for pointing out this inconsistency. While we had used a false discovery rate correction for multiple comparisons, that was not made clear in the initial submission. A clarifying section in the methods has been added to address this.

3. The correct way to test for sex differences is to run ANOVA with sex as a covariate. The authors should run 3-way ANOVA (Fig. S1) and 2-way ANOVA (Fig. S2) to correctly compare the effect of biological sex. If there is no main effect of sex or interaction between sex and treatment, only then should the authors claim no sex differences.

We appreciate the reviewer’s insight into the correct statistical test to run. The corresponding tables have been updated to reflect the proper statistical analyses. 

4. The number of mice used per sex/treatment is not powered enough to find sex differences. This limitation should be mentioned in the discussion.

We agree that the reviewer that this is a limitation in how our study has been done. We have added a paragraph in the discussion to address this limitation.

5. In line 113, remove the statement “implicating a sex-specific control over novel social interactions” since there’s no statistical significance.

This statement has been removed to reflect the statistical findings. We thank the reviewers for this comment.

6. In prior work from the lab, Newmyer et al. found reduced locomotion following the activation of IL/PL VIP neurons. Did the authors observe any difference in the total distance traveled?

This was a good point, and we have added overall locomotion data in both figure 3 and supplementary figure 3. No differences were seen in total distance traveled, in contrast to the Newmyer et al. manuscript. We postulate these differences are due to the gating mechanism that is explored in the discussion. We have expanded this section to address novelty-seeking in general, encompassing both palatable food intake as well as exploratory behavior as measured by overall locomotion.

7. In Fig. S2A, there are 7 female sham and 9 female caspase mice, but in S2B, there seem to be 8 female sham and 10 female caspase mice. Same issue with S2C. Please fix this and report the correct values.

We appreciate the reviewer for pointing this out. This was a simple clerical error and the figures and accompanying statistics have been corrected. While some statistics were altered, none resulted in any statistical conclusions being changed.

8. In Fig. S2, it looks like female caspase mice consumed more food than sham mice. Did the ablation of VIP neurons alter the body weight of mice? 

We acknowledge that these data would provide additional benefit and confidence that ablation of VIP neurons did not cause changes in food intake, but unfortunately the weights of the animals were not recorded at the time of euthanization. Based on observation however, caspase ablation had no overt effect on body weight. 

Minor comments:

1. Line 148 should say Gq-coupled DREADD; Line 156: it should be trials and not trails; Minor grammatical errors and spelling mistakes.

We appreciate the reviewer pointing out these mistakes. They have been appropriately corrected in the manuscript.

Reviewer #2: Hatter and Scott present a well-written manuscript assessing the behavioral effects of infralimbic cortex VIP ablation on impulsivity-like behavior and other mPFC-related tasks. They use a caspase virus with a transgenic strategy in VIP-IRES-Cre mice and combine male and female mice. VIP caspase-ablated mice showed increased impulsive-like behavior with the 12.5 intertrial interval, as assessed in the three-choice serial reaction time task (3CSRTT), with trends in the same direction at the shorter ITI’s and premature responses. There were no other stark differences between sham and VIP-ablated mice when they were further tested for social interaction with a novel mouse, free exploration in the Phenotyper box, and acute high-fat diet consumption. Importantly, these behaviors were performed during the dark phase. While this is a straight-forward set of studies with the same animals, improving the treatment of sex, increasing specificity of behavioral language, and including more methodological details would improve this manuscript.

Major comments:

1. These experiments collapse the data for male (n=5-6 per group) and female (n=7-9 per group) mice. While there are no significant differences in sex for any of the behaviors, this may be because of a low sample size, especially among the males. These behavioral differences may be driven by the females, as indicated in the Supplementary Table of p-values. If adding more animals is not possible, the authors could indicate the male vs female data points within the Figures. For example, having open vs closed circles, or circles versus triangles, etc. would distinguish the sexes within the same bar graphs. Another recommended strategy would be to provide the means and standard errors of the variables in the supplementary tables (plus p-values, as already given) for each sex. This treatment of sex would not only improve the manuscript, but be useful for future studies.

We appreciate this insight into the presentation of the data and have adjusted the graphs that collapse the sexes to show the males with open circles and the females with closed circles. 

2. I have some issue with anxiety-like behavior being assessed in the PhenoTyper. Traditionally the open field test is performed in larger, uncovered arenas, and the Noldus PhenoTyper is enclosed on all sides and a smaller area space. I would make it clear to readers that this set-up is not a traditional open field, reporting the smaller dimensions. I would also refrain from generalizing anxiety-like behavior since other tasks of anxiety-like behavior, such as an elevated plus maze, or light-dark test, were not performed. Again, I recommend being more specific with word choice. 

We acknowledge that our assay was not performed as a traditional open field, and have adjusted the language in the manuscript that we assessed spatial anxiety with the open field in the PhenoTyper. Additionally, we remain confident that the assay was able to produce spatial anxiety behavior, as there was a clear avoidance of the center by the animals. While the size of the arena may change the crossing behavior, we have the dynamic range to assess the primary question that we wished to address. 

3. Another area of overgeneralization is regarding the acute high fat diet feeding test. The authors present the experiment as assessing high calorie food intake, hedonic feeding, or eating behavior, but the protocol was a 30-minute test. Research on hedonic feeding may entail more chronic high fat protocols assessing binge-like consumption. Further, general eating behavior was not assessed, so adding more specific language, or discussing as a limitation, would more accurately reflect the results. 

We agree with the reviewer that this was an overgeneralization on our part in the initial manuscript. In order to reflect this, we have adjusted our language to indicate that we were testing palatable food intake, as our assay does not allow us to differentiate between hedonic and caloric driven feeding. We have also added an additional sentence in the results section to further explain that we hypothesize that this assay does primarily measure hedonic feeding mechanisms, due to the fact that the assays is performed at the end of the dark cycle, when the animals are fully sated. 

4. The Figures have poor resolution. In Figure 1A, one can hardly see the ROI box in the middle panels or see the text “DAPI.” In Figure 2A, the text labels are also illegible.

We appreciate the reviewer informing us of this issue. It has been resolved for this submission. 

Minor comments:

The methods are generally well-explained, but some minor methods details can be added, such as: 

1. Test order of experiments 

We agree that this is an important addition, and it has been clarified at the beginning of the 3CSRTT section. 

2. 3CSRTT training vs testing timeline

We apologize for any omissions and now have included additional information regarding the timeline of 3CSRTT training and testing in the methods section.

3. Equipment manufacturer for the 3CSRTT

We appreciate the reviewer for pointing out this omission. This information has been added to the methods section. 

4. Did mice encounter same-sex conspecifics

We have included this information in the methods section to reflect that the animals did encounter same-sex conspecifics during the novel social interaction assay.

5. Minor writing edits: The second to last line, “impulsive responding during longer trials” could end with “in the three-choice serial reaction time test.” 

Line 103, The header, “Ablation of VIP interneurons does not affect interest in novel stimuli” is too general in that only a novel conspecific was tested, and not separate tests of social interaction and novel object recognition task.

Line 148, I think the phrase, “Gq-coupled does not alter” could use “DREADD” or “chemogenetic stimulation” in the sentence.

We appreciate the reviewer’s improvements, and the appropriate sections have been adjusted to reflect their suggestions.

---

## [Decision Letter · Decision Letter 1]

11 May 2023

Selective ablation of VIP interneurons in the rodent prefrontal cortex results in increased impulsivity

PONE-D-22-34113R1

Dear Dr. Hatter,

We’re pleased to inform you that your manuscript has been judged scientifically suitable for publication and will be formally accepted for publication once it meets all outstanding technical requirements.

Kind regards,

Andrey E Ryabinin, Ph.D.

Academic Editor

PLOS ONE

Additional Editor Comments (optional):

Reviewers' comments:

Reviewer's Responses to Questions

**Comments to the Author**

1. If the authors have adequately addressed your comments raised in a previous round of review and you feel that this manuscript is now acceptable for publication, you may indicate that here to bypass the “Comments to the Author” section, enter your conflict of interest statement in the “Confidential to Editor” section, and submit your "Accept" recommendation.

Reviewer #1: All comments have been addressed

2. Is the manuscript technically sound, and do the data support the conclusions?

Reviewer #1: Yes

3. Has the statistical analysis been performed appropriately and rigorously? 

Reviewer #1: Yes

4. Have the authors made all data underlying the findings in their manuscript fully available?

Reviewer #1: (No Response)

5. Is the manuscript presented in an intelligible fashion and written in standard English?

Reviewer #1: (No Response)

6. Review Comments to the Author

Reviewer #1: (No Response)

7. PLOS authors have the option to publish the peer review history of their article (what does this mean?). If published, this will include your full peer review and any attached files.

Reviewer #1: No

---

## [Editor Report · Acceptance letter]

25 May 2023

PONE-D-22-34113R1 

Selective ablation of VIP interneurons in the rodent prefrontal cortex results in increased impulsivity 

Dear Dr. Hatter:

I'm pleased to inform you that your manuscript has been deemed suitable for publication in PLOS ONE. Congratulations! Your manuscript is now with our production department. 

Kind regards, 

on behalf of

Dr. Andrey E Ryabinin 

Academic Editor

PLOS ONE